# Race for the Cure: From the Oldest to the Newest Monoclonal Antibodies for Multiple Myeloma Treatment

**DOI:** 10.3390/biom12081146

**Published:** 2022-08-19

**Authors:** Gianfranco Lapietra, Francesca Fazio, Maria Teresa Petrucci

**Affiliations:** Hematology, Department of Translational and Precision Medicine, Sapienza University, 00161 Rome, Italy

**Keywords:** multiple myeloma, immune therapy, monoclonal antibodies, nanobodies

## Abstract

Multiple myeloma is characterized by a wide clinical heterogeneity due to an intricate network of interactions between bone marrow-resident clonal plasma cells and the microenvironment. Over the last years, dramatic improvement in the understanding of these pathways led to the introduction of novel drugs with immune-mediated mechanisms of action. Some of these compounds, such as the anti-cd38 daratumumab and isatuximab, the anti-slamf-7 elotuzumab, and the antibody-drug conjugate belantamab-mafodotin, have been tested in large clinical trials and have now fully entered the real-life management. The bispecific T-cell engagers are under investigation with promising results, and other satisfactory data is expected from the application of nanotechnologies. The perfect timing to introduce these drugs in the sequence of treatment and their adverse events represent new challenges to be addressed, and further experience is required to improve their use.

## 1. Introduction

Multiple myeloma (MM) is a plasma-cell disorder, accounting for 10% of hematologic malignancies, and it is characterized by a wide clinical heterogeneity due to an intricate network of interactions between bone marrow-resident clonal plasma cells (PCs) and the microenvironment.

Over the last decades, the use of therapeutic regimens mainly based on the combination of two different classes of drugs, i.e., immunomodulants (IMiDs) and proteasome inhibitors (PIs), led to dramatic improvements in outcomes, with significant elongation of overall-survival (OS) and progression-free survival (PFS), both in newly diagnosed (ND) and in relapsed-refractory (RR) patients. Nevertheless, MM remains an incurable disease with early mortality [1] or frequently relapsing, and less than 15% of affected subjects have a life expectancy as the normal population [2].

More recently, the anti-MM armamentarium has been enriched with different antibody compounds that are able to target specific clonal plasma cells antigens. These novel drugs showed safety and efficacy in the setting of relapsed/refractory MM are also proving useful in front-line approaches. These drugs act on multiple levels by interfering with pathogenic immune-mediated pathways.

### 1.1. Targets of Antibody Compounds

Any surface or intra-cytoplasmic antigen expressed by the aberrant plasma cell might be the potential target for antibody therapy. The ideal ligand should be a protein uniformly, abundantly, and exclusively produced by the tumor cells to increase the anti-MM effect and reduce the off-target side effects. Furthermore, the possibility of down-regulation for this protein should be very low to not impair the long-term efficacy of the novel drug.

To date, the most suitable and available plasma cell targets are CD38, SLAMF7 and BCMA.

-**CD38:** it is a transmembrane glycoprotein belonging to the family of cyclic ADP ribose hydrolases and is widely expressed on the surface of hematopoietic and non-hematopoietic cell lines, with greater intensity on both the non-clonal and the clonal PCs. CD38 plays multiple functions: as a receptor, it interacts with CD31 on the surface of T cells to produce a great variety of inflammatory cytokines; as an enzyme, it is involved in the regulation of cellular calcium trafficking [3]. Its central role in the oncogenesis of MM has been shown by different in vitro assays: the increase of levels of nicotinamide adenine dinucleotide (NAD+) due to its enzymatic activity seems to be associated with resistance to apoptosis [4] and the subsequent production of cyclic ADP ribose (cADPR) seems to ease the tumor escape from the immune system [5]. High expression of CD38 has also been associated with the formation of nanotubes transferring mitochondria from the stromal cells to myeloma cells to increase their proliferation and survival [6].-**SLAMF7:** also referred to as CS1, CD2 subset-1, CRACC, and CD319, it is a surface protein belonging to the signaling lymphocyte activation molecule (SLAM) family. SLAMF7 is highly expressed by non-clonal and clonal PCs but can also be found on the surface of CD8+ T cells, NK cells, and mature dendritic cells. Even if its function is not fully understood yet, SLAMF7 seems to play a key role in the interaction between the myeloma cells and the microenvironment: in particular, the stimulation of this SLAM protein might recruit the immune cells of the bone marrow niche to support the survival of PCs [7].-**BCMA:** also known as tumor necrosis factor receptor superfamily member 17 (TNFRSF17), it actually belongs to the TNF-receptors superfamily. This protein is consistently expressed on the surface of normal and malignant B-cells and by mature B cells and has a crucial role in the survival of long-lived plasma cells in the bone marrow. In particular, the binding of the ligand B-cell activating factor (BAFF) and a proliferation-inducing ligand (APRIL) to BCMA, expressed on the surface of clonal plasma cells, results in the activation of survival and proliferation pathways and in multidrug resistance [8]. BCMA can also be solubilized by y-secretase and interfere with circulating B-cell activating factor (BAFF), which is required for normal B-cell development. Therefore, serum BCMA is elevated in myeloma patients and directly correlates with tumor burden and outcome [9,10].

To date, several other targets are under investigation:-Surface molecules: GPRC5D, FcRH5, CD138, CD40, CD74;-Signalling molecules: IL-6, RANKL, BAFF, VEGF, DKK1;-Immune checkpoint inhibitors: PD1, PDL1, CTLA4, KIR.

### 1.2. Mechanisms of Action of Antibody Compounds: Overview

Antibody compounds against MM plasma cells might also be classified according to their mechanisms of action. The oldest drugs, such as daratumumab, isatuximab, and elotuzumab, are humanized IgG1-K monoclonal antibodies (moAbs): they kill tumor cells, especially via Fc-dependent immune effector mechanisms, including complement-dependent cytotoxicity (CDC), antibody-dependent cell-mediated cytotoxicity (ADCC) and antibody-dependent cellular phagocytosis (ADCP). Their action also involves the microenvironment cells through the activation of several mostly unknown pathways [11,12].

More recently, other types of antibodies have been engineered, such as antibody-drug conjugates (ADCs) and bispecific T-cell engagers (BITEs).

ADCs, such as belantamab-mafodotin and indatuximab ravtansine, are made up of an anti-tumor antibody chemically linked to a cytotoxic payload: using the antigen-antibody recognition, the injurious substance can directly be released into the myeloma-cells and concentrate within them to block the cycle [13].

BITEs are designed to simultaneously bind a T-effector cell and the myeloma cells. In particular, they are fusion proteins consisting of two single-chain variable fragments, which bind to CD3 on the surface of the T-cell and to a specific target on the surface of the tumor cell, respectively. Therefore, the compound induces the creation of a cytolytic synapsis, addressing the release of granzyme-B and perforin from the effector cell towards the aberrant plasma cell [14]. Several drugs belonging to this class are under investigation, such as talquetamab, teclistamab, and cevostamab, with promising results (Figure 1).

## 2. Starting from the Oldest: The “Naked” Monoclonal Antibodies

### 2.1. Anti-CD38: Daratumumab and Isatuximab

To date, Daratumumab is approved for patients with NDMM in combination with either PIs and IMiDs and for patients with RRMM, as a single agent or in association. Isatuximab is indicated only in patients with relapsed disease.

#### 2.1.1. Daratumumab: Mechanism of Action

Daratumumab is the first fully humanized anti-CD38 IgG1-K introduced in the treatment of MM. By interaction with CD38, it induces cell death through the above-mentioned immune mechanisms: CDC, ADCC, and ADCP. There is also consistent evidence of an off-target regulatory effect exerted by moAb on the microenvironment, with a reduction of T and B regulatory cells and an increase of NK and CD8+ cells [15].

#### 2.1.2. Daratumumab in Relapsed/Refractory Multiple Myeloma

Daratumumab was firstly approved as a single agent for the treatment of heavily pre-treated patients on the basis of two key trials: GEN501 (phase I/II) and SIRIUS (phase II) [16,17]. Both trials enrolled patients previously exposed to IMiDs and PIs and showed in this setting the efficacy and the safety of the anti-CD38. In particular, a pooled analysis of the two studies confirmed a satisfactory overall-response rate (ORR), with elongation of PFS and OS and a depth of response increasing in time if the drug continued to be administered [18]. Due to synergic action on the immune system, the efficacy of daratumumab is improved by combination with other anti-myeloma drugs, especially PIs and IMiDs, which seem to sensitize the PCs to the antibody-mediated lysis [19]. Based on the promising results of phase I/II trial NCT01615029 investigating the efficacy of daratumumab in association with lenalidomide and dexamethasone (Rd) [20], the POLLUX trial compared daratumumab in combination with Rd versus Rd in 569 patients previously treated with one or more lines of therapy [21]. At 12 months, the PFS rate was 83.2% in the daratumumab group versus 60.1% in the control group, and, after a follow-up of 13.5 months, progression of disease or death occurred in 53 patients in the daratumumab group versus 116 patients in the control arm, with a hazard ratio of 0.37 in favour of the first group. In spite of the higher incidence of neutropenia, diarrhoea, and infusion reactions in the experimental arm, the rate of grade 3 and grade 4 infections was not so different between the two groups, confirming the efficacy and safety of daratumumab combined with a regimen with IMiDs and high-dose steroid. Following the encouraging findings from a phase-1 trial evaluating both NDMM and RRMM patients, the association of daratumumab with other MM backbone treatments, including bortezomib and dexamethasone [22], the CASTOR trial was designed [23]. In the phase-3 CASTOR trial, 498 RRMM patients were randomized to receive treatment with only bortezomib and dexamethasone or with the addition of daratumumab. The 12-month rate of PFS was 60.7% in the experimental group and 26.9% in the control group. After a follow-up of 7.4 months, progression of disease or death occurred in 67 patients in the daratumumab group versus 122 in the control group. Thrombocytopenia and infusion-related reactions were the main side effects in the experimental arm, but none of them led to treatment discontinuation higher than in the control group. More recently, other combinations of daratumumab with IMiDs and PIs belonging to latter generations have been evaluated. In particular, in a phase 1 study, intravenous daratumumab plus pomalidomide and dexamethasone induced a very good partial response or better rate of 42% and was well tolerated in patients with heavily pretreated multiple myeloma [24]. In the ongoing phase 3 APOLLO trial, 304 RRMM pts were randomized to receive pomalidomide and dexamethasone alone or subcutaneous daratumumab plus pomalidomide and dexamethasone. These data show that subcutaneous daratumumab plus pomalidomide and dexamethasone was associated with significant progression-free survival benefit and deeper responses than pomalidomide and dexamethasone alone and represent a convenient treatment option for patients with relapsed or refractory multiple myeloma who have received at least one previous therapy, including lenalidomide and a proteasome inhibitor.

Currently, lenalidomide and bortezomib frontline exposure has raised a growing need for novel treatments for patients with relapsed or refractory multiple myeloma. A second-generation PI carfilzomib was firstly evaluated in a phase 1 study MMY1001 in combination with daratumumab and showed substantial efficacy with tolerable safety in relapsed or refractory multiple myeloma. In the randomized, multicenter, open-label, phase 3 study CANDOR trial, 466 patients were randomly assigned 2:1 to receive carfilzomib, dexamethasone, and daratumumab (KdD) or carfilzomib and dexamethasone (Kd). These data demonstrated that KdD resulted in a significant progression-free survival benefit compared with Kd. In addition, patients treated with KdD achieved improved overall and deeper responses across prespecified clinically important subgroups, including Lenalidomide exposed and lenalidomide-refractory subgroups. In addition, the observed adverse events associated with KdD were consistent with the known safety profiles of each agent, suggesting that the association with daratumumab does not result in additional toxicity [25].

#### 2.1.3. Daratumumab in Newly Diagnosed Multiple Myeloma

The proved efficacy of daratumumab in RRMM patients subsequently led to trials evaluating this moAb in front-line therapeutic combinations to maximize its potential both in transplant-eligible and transplant-ineligible patients.

The evidence of the efficacy of daratumumab in transplant-ineligible patients proceeds from the two main trials, ALCYONE and MAIA [26,27]. The first study randomized 706 naïve patients to receive the triplet VMP (bortezomib, melphalan, and prednisone) alone or in combination with the anti-CD38, while MAIA enrolled 737 people to compare Rd versus Dara-Rd. Both trials showed satisfying superior PFS and OS in the experimental arm compared to the control arm, with no significant differences regarding age and performance status. In spite of that, the high-risk cytogenetic aberrations as defined by the International Myeloma Working Group (IMWG) [del(17p), t(4;14), t(14:16)] still represented independent unfavourable prognostic factors. Lower respiratory tract infections arose as the most frequent side effects in people receiving daratumumab but did not impair the general outcome. The main adverse effect was represented by infections of the respiratory tract, but they were not a cause of discontinuation of treatment.

The first large trial about daratumumab-based combinations in transplant-eligible populations was CASSIOPEIA. This study enrolled 1085 patients, randomly assigned to pre-transplant induction and post-transplant consolidation with triplet VTD (bortezomib, thalidomide, dexamethasone) alone or in combination with daratumumab, followed by maintenance with daratumumab versus observation [28]. CASSIOPEIA met both primary end-points, showing a higher rate of stringent complete response according to IMWG criteria, and a longer PFS in the experimental arm; a longer follow-up will also bring in-depth assessments of OS. The efficacy of a four-drug daratumumab-based front-line regimen was later confirmed by the GRIFFIN trial: 207 patients received induction and consolidation with VRD or Dara-VRD, followed by maintenance with lenalidomide alone or in combination with the anti-CD38. In addition, in the transplant-eligible population, if on treatment with daratumumab, not-life threatening infections were frequently reported, and high-risk cytogenetics has continued to represent an independent prognostic factor.

To date, several trials have addressed both RRMM patients and transplant eligible/ineligible naïve population, and evaluation of the feasibility and efficacy of daratumumab combined with different classes of drugs is ongoing (NCT03993912, NCT03742297, NCT03652064, NCT03217812, NCT04052880, NCT04009109, NCT03695744, NCT02918331, MASTER). Their results are eagerly awaited to better clarify the role of the anti-CD38 in the anti-MM therapy.

#### 2.1.4. Daratumumab: Side Effects

Based on the results of the pivotal studies and on real-life experiences, the treatment with daratumumab seems to be associated with a slight major susceptibility to the infection due to neutropenia and the impairment of cellular immunity [29]. With regard to opportunistic viral infections, the ESCGICH (ESCMID Study Group for Infections in Compromised Hosts) recommends seasonal influenza vaccination and appropriate prophylaxis in Varicella-Zoster Virus seropositive patients [30]. In contrast, there is not yet a universal consensus about anti-bacterial prophylaxis in people receiving anti-CD38, even if grade ¾ pneumonia has been reported as a common adverse event in the safety analysis of data from five phase 3 trials with daratumumab [31]. An Italian expert panel has recently addressed this issue [32]. In the agreed position, they recommend prophylaxis against *Pneumocystis jirovecii* in all patients for the entire duration of treatment with the anti-CD38 if in association with a high-dose steroid, while fluoroquinolone prophylaxis is encouraged during front-line therapy including daratumumab only in patients at high risk for infection identified on the basis of past medical history, age, and tumor burden. Another frequently reported side effect is represented by infusion-related reactions, mainly occurring at the first infusion and usually presenting with mild-to-moderate symptoms such as nasal congestion, cough, throat irritation, chills, vomiting, and nausea. Serious adverse reactions with bronchospasm, dyspnea, laryngeal edema, pulmonary edema, and hypoxia have been only reported in a few cases [33]. Therefore, based on some studies (PAVO, COLUMBA), showing a better safety profile and a non-inferiority compared to the intra-venous formulation, the subcutaneous administration of daratumumab is possible and largely preferred by both clinicians and patients.

#### 2.1.5. Daratumumab: Recurrent Interferences with Laboratory Tests

Another ssue to be taken into account in patients on treatment with daratumumab is represented by recurrent interferences with laboratory tests, in particular blood compatibility tests and disease quantification. The interference with CD38 on the surface of red blood cells could generate positivity in the Coombs test and complicate the safe provision of blood products to people on treatment and by interference with measurement tests of monoclonal components. Actually, due to their proteic nature, moAbs can be detected both by serum electrophoresis and by immune-fixation electrophoresis, usually in the γ-region [34]. Consequently, they may cause underestimation of laboratory disease quantification, impairing response assessment according to IMWG criteria. In 2016, a daratumumab-specific immune-fixation electrophoresis reflex assay was developed using a murine anti-daratumumab antibody. This specific Ig is expected to bind and shift the migration pattern of anti-CD38, allowing quantification of the only M-protein in the γ-region [35]. Over the last years, other assays with higher sensitivity and specificity have been designed, most of them using mass spectrometry [36,37]. However, to date, a standardized method to eliminate the interference has not been identified, and a universal consensus is required to conform response assessment reports in patients on treatment with daratumumab and other similar drugs.

The concern regarding the possible delay in stem cell collection for eligible patients due to interference of daratumumab with CD34+ hematopoetic progenitor cells has been exceeded by the evidence of minimum expression of CD38 on the surface of the staminal compartment [38].

#### 2.1.6. Isatuximab: Mechanism of Action

Isatuximab is a novel immunoglobulin G1-K moAb that binds selectively to a specific epitope on CD38. A preclinical study demonstrates that isatuximab can target clonal plasma cells through a combination of several mechanisms, including CDC, ADCC, ADCP, and immune cell depletion or inhibition of immunosuppressive cells. In addition, isatuximab appears unique among anti-CD38 moAbs as it can also induce direct apoptosis without cross-linking [39]. Additional effects have been recently described: in particular, it might sensitize the T-cells against CD38+ cells, causing a long-lasting “in-vivo vaccination effect” [40].

#### 2.1.7. Isatuximab in Relapsed/Refractory Multiple Myeloma

Based on the results of preclinical models showing enhanced activity of isatuximab if combined with IMiDs or PIs [41], two large phase-3 trials investigated the efficacy of adding this anti-CD38 to either an IMiD or PI-based regimen in the setting of patients with RRMM.

In particular, in the phase III randomized, multicentre ICARIA, 307 patients previously exposed to more than two lines of therapies were randomly assigned to receive isatuximab in combination with Pd or Pd [42]. After a median follow-up of 11.6 months, the median PFS was 11.5 months in the experimental arm versus 6.5 months in the control arm. This advantage was confirmed in the updated analysis with a longer follow-up of 35.3 months [43] across all the subgroups, including those with high-risk cytogenetics according to IMWG criteria and, notably, those with aberrations of chromosome 1, regardless of other chromosomal abnormalities [44]. The benefit of adding the novel anti-CD38 to a pomalidomide-based regimen also translated to a high proportion of patients achieving deeper response and, finally, the improvement of median OS: after one year, OS rates in the experimental and control arms were 73% and 62%, respectively. Therefore, the ICARIA trial led to the approval of isatuximab combined with Pd in RRMM patients who have received ≥2 prior therapies, including IMiD and PI.

IKEMA enrolled 302 patients with the previous exposition to a median number of two lines of therapies and randomized them to treatment with carfilzomib-dexamethasone (Kd) or in combination with isatuximab [45]. After a median follow-up of 20.7 months, PFS was 35.7 months in the experimental arm versus 19.15 months in the control arm, and in this case, the benefit was consistent across all the subgroups, including those with high-risk cytogenetics, and translated into a deeper response. Even if the OS data are still immature, this trial favoured the approval of isatuximab-Kd as an option in the treatment of patients with RRMM, previously exposed to ≥1 line of therapy. Isatuximab could also be a feasible and safe therapeutic option in patients with prior exposure to IMiDs and PIs either as a single agent or in combination with dexamethasone. A phase-2 trial evaluated both alternatives in 164 RRMM patients who were refractory at least to an IMiD and a PI or with prior exposure to ≥3 lines incorporating an IMiD and a PI [46]. The study showed satisfactory ORR, with the most benefit coming from the addition of steroids.

#### 2.1.8. Isatuximab: Side Effects

ICARIA and IKEMA showed that isatuximab-based regimens are not only effective but also safe solutions in the heavily pre-treated MM population. In the experimental arms of both trials, some of the main reported side effects are similar to those regarding daratumumab: the antibody nature might over-rate serum M-protein detection during disease evaluation and might interfere with CD38 on the surface of red blood cells, causing false positive Coombs test. In addition, most patients presented infusion-related reactions, which could be prevented with adequate prophylaxis. Other specific safety concerns are neutropenia in combination with Pd and hypertension in combination with Kd, but, to date, none of these adverse events has been related to an increased rate of mortality.

#### 2.1.9. Felzartamab

Felzartamab, also referred to as MOR202, is another novel moAb directed against CD38. It differs from daratumumab and isatuximab for the light chain, being λ, and for the mechanism of action [47]. Actually, this drug induces the death of plasma cells only by ADCC and ADCP, without recruiting the complement. Its efficacy has been evaluated in a phase I/II trial enrolling 91 patients with RRMM [48]. The study design included dose-escalation and dose-expansion of the drug as a single agent and in combination with other regimens (dexamethasone, Pd, Rd)

### 2.2. Anti-Slamf7: Elotuzumab

Elotuzumab has been approved for the treatment of RRMM in combination with IMiDs and high-dose steroids; in particular, it can be used associated with lenalidomide in patients previously exposed to a therapeutic line or to pomalidomide in those formerly treated with two lines.

#### 2.2.1. Elotuzumab: Mechanism of Action

Elotuzumab is a fully humanized IgG-k1 moAb specifically targeting the protein SLAMF-7. As mentioned above, this signaling molecule is expressed not only on the surface of PCs but also by NK cells. Therefore, elotuzumab exerts its action by mediating the killing of the PCs through recruiting NK cells (ADCC); some in vitro samples actually showed that this moAb is effective only when MM-cells are incubated with NK cells, and its function can be prevented by blocking CD16 [12]. This particular mechanism of action might explain the better results of anti-SLAMF-7 in combination with PIs or IMiDs rather than alone, as shown by different phase I/II trials [49,50,51].

#### 2.2.2. Elotuzumab in Relapsed/Refractory Multiple Myeloma

The promising results of these studies led to larger trials evaluating elotuzumab-based combinations. In particular, ELOQUENT-2 was a phase-3 trial that randomized 646 patients with RRMM to receive Rd or elotuzumab-Rd [52]. The study showed statistically significant differences regarding ORR and PFS in favour of the experimental arm, with a benefit being consistent across all the subgroups, including those with high-risk cytogenetics [del(17p), t(4;14)], even if it was major in patients with non-aggressive relapse. The final OS analysis, after a minimum follow-up of 70.6 months, confirmed the efficacy of the combination with anti-SLAMF-7: the patients enrolled in the elotuzumab-Rd arm presented a median OS of 48.3 months versus 39.6 months of those treated with Rd [53]. More recently, the phase-2 trial ELOQUENT-3 showed elotuzumab to be effective also in the setting of patients who were refractory or relapsed after lenalidomide [54]. One-hundred seventeen patients were randomly assigned to receive pomalidomide-dexamethasone (Pd) alone or in combination with anti-SLAMF-7. After a minimum follow-up of 9.1 months, the median PFS was 10.7 months in the experimental arm and 4.3 months in the control arm, with an ORR of 53% versus 26%. Many ongoing trials are evaluating the combination of elotuzumab with other drugs, including PIs (NCT03155100, Silvennoinen R, Helsinki University Central Hospital, Finland) and check-point inhibitors (NCT02726581, Bristol-Myers Squibb, United States of America and Europe), with the results eagerly awaited.

#### 2.2.3. Elotuzumab in Newly Diagnosed Multiple Myeloma

To date, elotuzumab is not approved for front-line therapy, and some studies are evaluating this use for both transplant-eligible and ineligible patients. The phase-3 trial ELOQUENT-1 compared Rd versus elotuzumab-Rd in unfit NDMM patients, based on the results of a similar Japanese phase-2 study that showed a better ORR in the experimental arm versus the control arm [55]. However, recently published data from ELOQUENT-1 did not confirm a statistically significant improvement in PFS in the 371 patients receiving elotuzumab-Rd versus 339 on treatment only with Rd at a median follow-up of 65.3 months [56]. On the other side, the efficacy and feasibility of induction with anti-SLAMF-7-based combinations for fit NDMM are the main endpoints of ongoing trials [(NCT02495922, Goldschmidt H, University Hospital Heidelberg, Germany), (NCT02969837, Jakubowiak A, University of Chicago, United States of America), (NCT02375555, Laubach JP, Dana-Farber Cancer Institute, United States of America)].

#### 2.2.4. Elotuzumab: Side Effects

Data from the ELOQUENT trials confirmed the high tolerability of elotuzumab, with a very advantageous safety profile. The reported cases of lymphopenia might be associated with a slightly higher incidence of infection in the experimental arms, even if treatment has never been discontinued. As previously reported for daratumumab, also during the first infusion of anti-SLAMF-7, allergic reactions were reported, mainly with mild-to-moderate symptoms. Nevertheless, premedication is highly recommended.

“Naked” moAbs, with their target and their indication, are summarized in Table 1.

## 3. Arriving to the Newest: The “Combined” Monoclonal Antibodies

### 3.1. Antibody-Drug Conjugates: Belantamab-Mafodotin, AMG224, MEDI2228, HDP-101

Belantamab-mafodotin has been recently approved as a single agent for heavily pre-treated patients with RRMM, already exposed to IMiDs, PIs, and anti-CD38 moAbs.

The other similar drugs are still under investigation.

#### 3.1.1. Belantamab-Mafodotin: Mechanism of Action

Belantamab-mafodotin is the first-in-class fully humanized anti-BCMA IgG1, linked to the microtubule-disrupting agent monomethyl-auristatin-F. Following interaction with BCMA on the surface of aberrant plasma cells, the drug is immediately internalized and releases the cytotoxic substance. By interference with the mitotic spindle, this agent blocks cell growth and induces apoptosis [57]. The first tests performed in 2014 showed that belantamab-mafodotin kills the MM cells not only via ADCC but also via macrophage-mediated phagocytosis: actually, the administration of this novel molecule into the long bone of mouse models was significantly associated with increased macrophage infiltration. All death-markers released by cells targeted with belantamab promote dendritic cell activation, potentially inducing a long-lasting adaptive immune response against the tumor [58]. Therefore, belantamab-mafodotin drives immunogenic cell death, being responsible for novel modalities of MM suppression.

#### 3.1.2. Belantamab-Mafodotin for Relapsed/Refractory Multiple Myeloma

Based on the promising results of a phase 1 clinical trial, showing single-agent activity of belantamab in heavily pre-treated patients, including those with previous exposure to daratumumab [59], the phase 2 study DREAMM-2 has been performed all over the world to grant the approval of this ADC by authorities. This trial randomized 223 patients with RRMM to receive either belantamab 2.5 mg/kg every three weeks (cohort A) or 3.4 mg/kg every three weeks (cohort B) until death or progression. At data cut-off, 196 patients were included in the intention-to-treat analysis [60]. It confirmed an OR in more than 30% of patients in both arms, with 20% achieving at least a very good partial response (VGPR) and a probability of having a duration of response of 4 months or longer being estimated at around 78% in the cohort A and 87% in cohort B. At the median duration of follow-up of 6.3 months in the first group and 6.9 months in the second group, the median PFS was 2.9 months and 4.9 months, respectively. Considering the better safety profile, with the same anti-myeloma activity, the lower dose of belantamab was the recommended one. Therefore, the update after 13 months of follow-up regarding patients on treatment with belantamab 2.5 mg/kg has been recently published [61]. OR > 30% was confirmed, with an improvement of deepness during treatment. The median PFS and the median OS were 2.8 months and 13.7 months, respectively, with the maximum benefit in those achieving a deeper response, regardless of the cytogenetic risk and of renal impairment.

The quick approval of belantamab led to many other trials aimed at investigating the role of this drug in different phases of the disease. The phase I/II DREAMM-6 has shown efficacy and safety of belantamab combined with either bortezomib + dexamethasone (Vd) or Rd in patients with RRMM exposed, at least to a prior line of therapy [62]. For the same setting of patients, other combinations being explored are carfilzomib (NCT05060627, López-Carrero C, Maldonado R, Spain), cyclophosphamide (NCT04822337, Atrash S, Wake Forest University Health Sciences, United States of America), pomalidomide (NCT04484623, GlaxoSmithKline, United States of America, Europe, Asia, Australia), and novel agents, such as gamma-secretase inhibitors and immune checkpoint inhibitors, including in the ongoing phase I/II trials DREAMM-5 [63]. The feasibility of maintenance with belantamab in heavily pre-treated patients either after salvage ASCT or after CAR-T infusion will be investigated by the trials NCT05065047 and NCT05117008, respectively. As front-line therapy, this novel ADC is being evaluated in combination with VRd for transplant-eligible patients (NCT04802356, López-Carrero C, Maldonado R, Spain) and with Rd in those who could not benefit from ASCT (NCT04808037, Dimopoulos MA, National and Kapodistrian University of Athens School of Medicine, Greece).

#### 3.1.3. Belantamab-Mafodotin: Side Effects

The trials investigating the novel ADC showed a favourable safety profile, with main side effects being similar to those reported in other moAbs. In particular, in the DREAMM-2, thrombocytopenia has been the recurrent form of hematologic toxicity, followed by anaemia, in both arms, with a slight preponderance in the 3.4 mg/kg group. Infusion-related reactions were mild-to-moderate, reported in less than 20% of enrolled patients. Nevertheless, the main safety concern is corneal toxicity. Actually, corneal events were diagnosed in 69% of the population treated with belantamab in DREAMM-1 [59] and also represented a frequent side effect in both arms of DREAMM-2, with grade 1-2 keratopathy being reported in 43% and 54% of 2.5 mg/kg and 3.4 mg/kg cohorts, respectively [60]. A review of corneal examination findings from patients in DREAMM-2 showed micro cyst-like epithelial changes (MECs) in 72% of cases, mainly causing blurred vision and dry eye [64]. These symptoms were associated with a need for dose delays or reductions in most cases, and, to date, no cases of permanent vision loss have been reported. Even if the pathogenic mechanism of corneal toxicity is still unclear, it can be due to the internalisation of monomethyl-auristatin-F by corneal limbal epithelial cells, leading to their apoptosis: whether the drug reaches the ocular anterior chamber by tear film or by bloodstream is still highly debated. Corneal examination by slit lamp and visual-acuity assessment before and during every treatment cycle should be performed to detect any change from baseline, with a case-by-case evaluation of dose delays or dose modifications [65]. As confirmed by a sub-analysis of DREAMM-2, there is no need for prophylactic use of steroid eye drops.

#### 3.1.4. Other Antibody-Drug Conjugates Targeting BCMA

Many other ADCs targeting BCMA are being investigated, representing a therapeutic chance, especially for heavily pre-treated patients with RRMM.

-AMG224 is a fully humanized IgG1 antibody directed against BCMA and conjugated to a maytansinoid, a derivative of a cytotoxic agent isolated from plants of the genus *Maytenus*. Its activity against MM has been evaluated in a phase I trial, which enrolled 42 patients with a median number of 7 prior lines of therapy: 29 received the drug in the dose-escalation phase and 11 in the dose-expansion [66]. In both phases, manageable hematologic side effects and grade 1-2 infusion-related reactions were reported. In addition, in this case, the treatment was associated with the onset of keratopathy requiring dose delays or dose modifications, with no need for discontinuation. The ORR was 23%, with a median duration of response of 14.7 months reported in the second cohort.-MEDI2228 is another anti-BCMA conjugated to a pyrrolobenzodiazepine with tumor activity due to its ability to link DNA and induce fatal damage. Based on preclinical studies showing efficacy against MM-cells, especially if combined with PIs [67], a phase I trial has investigated its use in 82 patients with RRMM [68]. The ORR was 61% with acceptable hematologic toxicities and no reported cases of keratopathy, even if 58.5% of patients experienced photophobia; in 41.5% of patients, this was grade 1 or 2, while it was grade 3 or 4 in 17.1%. In addition, considering the high rate of patients with prior exposure to daratumumab enrolled in this trial, MEDI2228 might be a worthy alternative for those relapsing after therapy with the “naked” moAbs, and further studies to confirm this observation are warranted.-HDP-101 differs from the other ADCs directed against BCMA since the payload is represented by α-amantin, a derivative of mushrooms genus *Amanita*, interfering with protein biosynthesis through inhibition of RNA polymerase II subunit A. Pre-clinical studies showed that this drug induces immunogenic cell death in myeloma cell lines, with a particular affinity towards those harbouring del17p, suggesting the possibility of using this high-risk cytogenetic aberration as a potential target in future [69].

### 3.2. Bispecific Antibodies

These novel drugs are under investigation for RRMM, and none of them has been approved yet. As mentioned above, they are designed with a dual antigen specificity to create a cytolitic synapsis between the patient′s own T cells and the tumor cells. In particular, in the last decade, two different constructs with this aim have been engineered: BiTEs (bispecific T-cell engagers) and DuoBody [70]. BiTEs are fusion proteins consisting of two single-chain variable fragments, connected by a flexible linker and binding to CD3 on the surface of the T cell and to a specific target on the surface of the tumor cell, respectively. DuoBody products are generated through the mechanism of “Fab-arm exchange”: this is a sequence of artificially induced mutations and recombination involving heavy and light chain homodimeric fragments from two different moAbs, targeting the CD3 and the antigen on the surface of aberrant plasma cells, respectively. This manipulation leads to the creation of a single heterodimeric, bispecific antibody that is supposed to be characterized by major stability. To date, only BiTEs have been used in the MM therapeutic research.

#### 3.2.1. BiTEs Targeting BCMA

Due to its high expression on the surface of aberrant plasma cells, BCMA also represents a valid target for BiTEs. Therefore, different trials are currently investigating these constructs.

-Teclistamab (JNJ-64007957): it is a humanized IgG4 binding simultaneously BCMA on the surface of MM cells and CD3 on the surface of T cells. In vitro assays showed that this drug induces cytotoxicity of BCMA+ cells through multiple mechanisms, including dose-dependent lysis, T-cell activation, and cytokine release, especially if in the presence of a γ-secretase inhibitor [71]. These observations have been confirmed in murine models and in ex vivo assays performed on bone marrow samples from MM patients. Based on these encouraging results, the safety and efficacy of teclistamab are being evaluated in the ongoing trial MajesTEC-1 (NCT03145181, Dimopoulos MA, National and Kapodistrian University of Athens School of Medicine, Greece). MejesTEC-1 is an open-label, single-arm, phase I trial in which teclistamab is administered intravenously or subcutaneously in different cohorts of RRMM patients with step-up dosing. To the last cut-off analysis, overall, 157 patients, with a median of six previous therapy lines, received at least one dose of the novel drug [72]. Forty of them were administered the recommended phase 2 dose (once per week 1500 μg/kg by subcutaneous route). In this last group, after a median follow-up of 6.1 months, the ORR was 65% and confirmed across all the subgroups, including 33 triple-class refractory patients. These responses were durable and deepened over time, with six evaluable MRD patients achieving MRD-negative complete response, evaluated by immunoglobulin gene rearrangement sequencing.-AMG420, formerly known as BI836909, is another BiTE targeting BCMA and inducing MM cell death through the same mechanism as teclistamab. Actually, studies performed in vitro on BCMA+ cell lines and in vivo on murine models and ex vivo assays on bone marrow samples from MM patients showed that this drug selectively mediates apoptosis of BCMA+ cells by recruiting T-cells [73]. NCT03836053 is the first-in-human, phase 1, a dose-escalation trial evaluating this novel drug [74]. It enrolled 42 patients with RRMM and a median of 5 prior therapy lines, including daratumumab. Extramedullary disease and prior allogenic transplant were exclusion criteria, representing a limit in the evaluation of the efficacy of AMG420 in the setting of patients with very aggressive diseases. After a median follow-up of 8 months, the ORR was 31% but at the maximum tolerated dose of 400 μg/die, the response rate increased to 70%. In the group exposed to this dose, MRD negativity was attained in 71% of cases even if it is evaluated per protocol by flow-cytometry.-AMG701 is a half-life extended anti-BCMA BiTE, formerly tested in cynomolgus monkeys in which it depleted aberrant plasma cells [75]. In further studies performed in MM cell lines and in autologous cells from RRMM patients, it showed to induce T-cell-dependent cellular cytotoxicity, especially if in combination with lenalidomide and pomalidomide [76]. The synergic effect of AMG701 with IMiDs might overcome the pro-tumor effect due to the microenvironment. Based on these results, the ongoing trial NCT03287908 is investigating in heavily pre-treated patients AMG701 monotherapy to assess the recommended phase 2 dose, to be also used in combination with pomalidomide. The initial results of this study showed in a population of 71 patients an ORR of 36%, increasing to 83% in those who underwent early escalation [77].-REGN5458 is a fully human BiTE with proven efficacy in vitro and in animal models, especially if combined with checkpoint inhibitors [78]. In these in vivo models, it showed an anti-tumor activity with faster kinetics than that of anti-BCMA CAR-T constructs injected in similar animals, suggesting the potential use of this novel drug to debulk very aggressive disease. NCT03761108 is the first-in-human trial aimed at establishing the safety and tolerability of REGN5458 in patients with RRMM. As of data cut-off, 68 patients, with the majority being penta-refractory, were treated in the dose-escalation cohort with full doses ranging from 3 to 400 mg [79]. The highest response rate was 73.7% and was observed among patients treated at 96 and 200 mg dose levels, with acceptable safety and tolerability profile.-CC-93269 is a BiTE characterized by a bivalent binding to BCMA and is under investigation in the trial NCT03486067 for patients with RRMM with no prior exposure to anti-BCMA therapy. The interim results on 30 patients showed an ORR of 43.3% with a complete response/stringent complete response of 16.7% and MRD-negativity achieved in the majority of responders [80].-Elranatamab, also known as PF-06863135 or PF-3135, is a humanized IgG2 anti-BCMA BiTE. Based on promising results of MagnetisMM-1, a phase 1 trial investigating efficacy and safety of elranatamab alone and in combination with IMiDs in RRMM, the phase-2 trial MagnetisMM-3 will evaluate this novel drug as a single agent in patients who should be at least triple refractory, also including those with prior exposure to other anti-BCMA [81].-TNB-383B is a unique anti-BCMA with major activity on effector T-cells more than on regulatory T cells if compared to similar drugs. The first-in-human, phase 1, dose-escalation/expansion study in RRMM with prior exposure to ≥3 lines showed an ORR of 79%, also confirmed in triple-refractory patients, again showing the efficacy of this class of drugs, especially in heavily pre-treated patients [82].

#### 3.2.2. BiTEes Targeting BCMA: Side Effects

The anti-BCMA BiTEs showed high rates of safety and tolerability across all the above-mentioned trials. Medullar toxicity was frequently reported, with anaemia followed by neutropenia and thrombocytopenia as the main side effects. Subsequent infections, including a significant rate of grade-3 events, were especially observed in patients treated with AMG420, REGN5458, and CC-93269, even if none of them led to discontinuing the treatment [74,78,80]. Among the non-haematological adverse events, cytokine-release syndrome (CRS) was the most frequently reported with all the drugs. CRS is potentially life-threatening toxicity due to high-level immune activation with the release of inflammatory cytokines, such as IL-6 and interferon-γ, from activated lymphocytes and myeloid cells [83]. The clinical presentation can greatly vary, ranging from mild reactions with fever, nausea, and myalgia to serious manifestations, with desaturation and hypotension. The majority of cases of CRS recorded in the trials about anti-BCMA BiTES were grade 1-2, being treated only with vigilant supportive care, usually occurred within 2 days from the infusion, and did not cause fatal events, confirming again the safety of these novel drugs.

#### 3.2.3. BiTEs Non-Targeting BCMA

Even if targeting BCMA, both by using BiTEs (see above) and by using CAR-T [84,85], has shown encouraging results in the therapy of RRMM, the expression of this antigen on the surface of aberrant plasma cells may vary over time, due to action of γ-secretase and to therapeutic pressure. This down-regulation could be responsible, for instance, for relapses reported either in patients after anti-BCMA CAR-T therapy [86,87] or after exposure to BiTEs [88]. The mechanisms of escape are several and unclear, even if heterozygous deletions of the BCMA gene detected in high percentages of MM patients may represent a risk factor [87,88]. Therefore, there is an emergent need for the identification of additional targets. Recent studies going in this direction have been investigating GPRC5D and FcRH5.

#### 3.2.4. BiTEs Non-Targeting BCMA: Anti-GPRC5D

Orphan G protein-coupled receptor, class C group 5-member D (GPRC5D), formerly identified in hair follicles, has been later found also on the surface of CD138+ MM cells, with a distribution that is similar to that of BCMA, even if its expression is independent [89]. Therefore, the novel BiTE talquetamab directed against GPRC5D has been recently designed [90]. In vitro and ex vivo experiments showed that this drug induces tumor lysis by T-cell activation, even in samples from patients with high-risk cytogenetics and prior exposure to anti-CD38 moAb. The efficacy of talquetamab is directly related to the level of expression of the target on the surface of MM cells and inversely proportional to the amount of regulatory T-lymphocytes in the bone marrow niche. Furthermore, these assays also showed synergic effects between this BiTE and daratumumab or pomalidomide. Based on these results, the phase 1 dose escalation/dose expansion trial MonumenTAL-1 is evaluating the tolerability and safety of this drug in a cohort of patients with RRMM. To the last data cut-off, among 95 highly pretreated patients, including those with prior exposure to anti-BCMA, the 0RR was 70%, with manageable CRS, grade ¾ neutropenia, and skin toxicity being reported as the main adverse events [91].

#### 3.2.5. BiTEs Non-Targeting BCMA: Anti-FCRH5

Fc receptor-homolog 5 (FcRH5) is a surface antigen belonging to the immunoglobulin-superfamily and restricted to mature B-cells, including plasma cells. Even if its function is unknown, its expression is higher in MM cells [92]. Therefore, the trial NCT03275103 is evaluating the use of cevostamab, a novel BiTE targeting FcRH5, in patients with RRMM. Preliminary data from a cohort of 160 patients, including those previously treated with CAR-T, other BiTEs, and ADC, showed encouraging results: a dose-dependent increase of ORR is associated with an acceptable toxicity profile, similar to that of other BiTEs [93].

“New” moAbs, with their target and their indication, are summarized in Table 2.

### 3.3. Other Antibodies: Overview

The significant improvement in understanding the biology of multiple myeloma and complex mechanisms underlying refractoriness and relapses led to a plethora of investigations to identify antibody compounds with tumor activity.

Beyond the above-mentioned molecules, other drugs with a certain efficacy against multiple myeloma are the following:-Indatuximab-ravtansine: also mentioned to a BT062, it is an anti-CD138 ADC. A phase I/II A study showed promising results in 35 heavily pre-treated patients, with ORR > 75% and mild-to-moderate gastro-enteric toxicity, but further investigations are needed to confirm them [94].-Siltuximab: it is an antibody targeting IL-6 that is involved in the growth of MM [95]. While its efficacy has been confirmed in other hematologic malignancies (i.e., Castelman disease), its use in MM is controversial. Even if it seems that it could delay progression from smoldering multiple myeloma to symptomatic forms [96], more evidence is required.-Denosumab: it is an inhibitor of RANK-L, which plays a leading role in the survival of osteoclasts. It might be of interest in the treatment of MM since it seems to slow progression through suppression of tumor escape in patients already on supportive treatment with this drug [97]. To date, in fact, this drug is approved in the setting of MM patients only to prevent disease-related bone fractures.

## 4. Overcoming the Newest: From the Microscopic to the Nanoscopic Therapeutic Approach

Although the use of moAbs has ensured the possibility of achieving extraordinary results in the challenging treatment of MM, some critical issues related to this immune therapy should be addressed.

First, despite evolution from murine to chimeric products, there may be a non-negligible risk of immunogenic response in the patients on treatment, with subsequent neutralization of the drug [98]. Another problem to be taken into consideration is represented by the size of moAbs: since they are macromolecules (molecular mass of 150 kDa), their penetration into the tumor site may be reduced, with impairment of efficacy. Paradoxically, drugs with a major affinity to their target could be most affected by this phenomenon due to their strong interaction with “peripheral” antigens [99]. If we consider that MM is a tumor arising from a hematopoietic niche, with the protection of the microenvironment, several physical and chemical barriers have to be penetrated before arriving at the core. Last but not least, moAbs preparation on a large scale requires high manufacturing costs, burdening the national health care systems.

In this context, the introduction of nanoscopic technologies could represent a good strategy to solve the limitations of approved cellular therapy.

### 4.1. The Nanobodies: Overview

In particular, in the onco-hematologic field, investigations regarding the so-called nanobodies have shown encouraging results. Nanobodies are heavy-chain-only antibodies lacking light chains and the first constant domain of the heavy chain, originally identified in the serum of camelids [100]. Compared to the classical moAbs, they are expected to be less immunogenic due to little size (molecular mass of 15 kDa) and the high grade of homology with the variable domain of human Ig (Figure 2a). This observation has been recently proved by immune assays performed on samples from 20 patients exposed to a nanobody on trial as a reagent for PET imaging of breast cancer [101]. Three months after administration, anti-drug antibodies were detected only in one case, without any effects on safety and pharmacokinetics. The small size of these products is also responsible for a major solubility, together with a high concentration of hydrophilic amino acids in the variable domain [102]. These biophysical properties may favour a major penetration into tissues and increase the possibility of targeting deep undetectable epitopes, with potential extraordinary consequences for eradication of MRD and the treatment of relapses in sanctuary sites, such as the central nervous system and gonads. The elevated solubility is also responsible for quick renal clearance, with a reduction of side effects due to accumulation [100]. Another advantage of nanobodies is represented by conformational stability due to disulphide bonds in the antigen-binding regions [103]. Therefore, while classical antibodies are usually administered intravenously or subcutaneously, nanobodies could be taken orally, increasing the patient’s adherence to treatment. Moreover, their monomeric structure could imply a reduction of manufacturing costs compared to the factory of classical moAbs.

### 4.2. The Nanobodies: Experimental Combinations with Radionuclides and Nanoparticles

Since nanobodies have been recently introduced, little is known about the efficacy and feasibility of their use in MM. Most investigations have been performed in vitro or in murine models in the last two years and have been addressed to target chemo-resistant dormant clones, being MRD eradication the most urgent unmet need for myeloma patients. Nanotechnology has been combined with radiotherapy to meet this aim. Actually, radionuclides can easily target residual pathologic cells, and, among these, α-particles are the ideal candidates, given their short range of energy deposition [104]. Ideally, any antibody against any target on the surface of myeloma cells can be coupled to the radionuclide. In 2020, a North American team showed that α-particles combined with daratumumab reduce tumor growth in an animal model [105], but due to the possible down-modulation of CD38 on the surface of refractory cells, novel deep epitopes targetable by nanobodies have been recently investigated. Promising results come from the use of nanobodies against SLAMF-7 combined with Actnium-225 in mice models of RR disease [106]: in all the animals, the targeted radionuclide therapy was actively uptaken by disease sites, as demonstrated in biodistribution studies with single-photon emission computed tomography, and this effect translated into increased survival and shift from a pro-tumor to an anti-tumor microenvironment. Based on the evidence that paraprotein can be anchored on the surface of myeloma cells [107], another target of nanobodies could be the complementarity-determining region of membrane immunoglobulin, which is specific for each patient and represents the so-called idiotype. A Belgian team recently worked on the design of radionuclides combined with nanobodies against the idiotype identified in five patients with NDMM, and in two of them, the product was successfully completed [104]. 

Therefore, future perspectives are several and exciting. Actually, nanobodies might be used not only as carriers of targeted radionuclide therapy but also as nanoparticles. These are organic or inorganic particles with a diameter ranging from 1 to 10 nanometres and are able to induce biophysical stress within the cells they are delivered in. Several materials have been investigated in the last years to find the ideal candidate. Among these, lipidic nanoparticles specifically evoking cytotoxic T lymphocytes via BCMA showed activation and long-term maintenance of immune response in in vitro MM models [108]. In addition, bortezomib-loaded metallic nanoparticles could be effective, either by selectively inducing death in MM cell lines or by damaging the microenvironment [109]. The down-modulation of pro-tumor phenotype of the microenvironment can also be favoured by lipidic nanoparticles loaded with inhibitors of Rho kinase, involved in the tumor migration [110].

The benefits coming from nanobodies could also be exploited for the generation of small-sized BiTEs. These constructs are expected to be more feasible than the macromolecular counterpart for the myeloma treatment in the advanced stage of the disease but, to date, only a few data are available and concern light T-cell engagers for solid malignancies with the expression of epidermal growth factor receptor [111].

Further in vivo investigations on larger animal and then human samples are required to confirm the efficacy and toxicity of these novel therapeutic approaches. The main safety concern of nanobodies is quick renal clearance. On a side, this pharmacokinetics could require a frequent dosage, with subsequent low adherence to the treatment; on the other side, it could worsen the renal function, with further impairment of the organ damage in patients with advanced stage disease. Instead, inorganic nanoparticles could accumulate into target organs, inducing the production of reactive oxygen species and genotoxicity.

### 4.3. The Aptamers

Beyond nanobodies, additional nanoscopic products to be explored for optimization of myeloma treatment are the aptamers. Aptamers are single-stranded oligonucleotides with a three-dimensional structure giving high affinity towards a specific target (Figure 2b). They are usually isolated by enrichment of single-stranded oligonucleotide wide libraries and reproduced through conventional DNA-synthesis techniques [112]. Like nanobodies, aptamers are characterized by low immunogenicity, high penetration capacity/high selectivity towards targets, and major solubility. Unlike nanobodies, their use is not impaired by the batch-to-batch variability affecting the antibody drugs in general. To date, only a few aptamers have proven useful in targeting proteins on the surface of aberrant plasma cell lines, such as TY04, annexin A2-binding aptamer, SL1 against tyrosine-protein kinase Met, and, more recently, apt69.T against BCMA [113,114,115,116]. In this case, too, further interdisciplinary efforts are necessary to move this innovative, sophisticated approach from bench to bedside.

## 5. Discussion: At the Finish Line or Still in the Race?

MM is a very complex disease characterized by sophisticated biogenetic mechanisms regulating the competition between different sub-clones and their evolution in space and in time [117]. This molecular heterogeneity is responsible for the great variety of clinical manifestations and for the discrepancies among outcomes.

Over the last years, the dramatic improvement in the knowledge of the landscape feeding MM has led to the identification of pathogenic targets and, therefore, to the experimentation of novel drugs. Some of these compounds, such as naked anti-CD38 and anti-SLAMF-7 moAbs and the ADC belantamab-mafodotin, showed their efficacy in wide clinical trials and have now fully entered real-life practice. Other drugs, such as the BiTEs, are still under investigation, even if the preliminary data are encouraging.

All the therapeutic antibodies mentioned above are the cause for a paradigm shift in the approach both to ND and to RRMM. This revolution brings novel challenges to be addressed.

First of all, there is concern regarding the emerging toxicities, such as keratopathy in patients treated with ADC and CRS in those receiving BiTEs. These side effects might be unusual to the haematologist and require close collaboration with other physicians to be faced.

The other great question is about the perfect timing to introduce the novel drugs into the sequence of MM treatment. Except for daratumumab, cellular therapy is now mainly used in late-stage RRMM. Actually, cellular therapy has been mainly investigated in the setting of RRMM patients based on the most urgent current clinical needs. However, referring to some pre-clinical evidence, the major benefit from antibody drugs could be achieved if used in earlier stages of the disease. The evolution of MM is accompanied by a progressive impairment of immune competence due to multiple mechanisms. In addition to the well-known immune-paresis, MM supports a tumor-permissive microenvironment. The immune-tolerant niche is characterized by increased expression of CD4+ Foxp3+ regulatory T-lymphocytes via interferon signalling on one side and by anergy of CD8+ cytotoxic T-lymphocytes via PD1-PDL1 signalling on the other side [97,118]. Furthermore, there is increased evidence of age-related immune disruption induced by standard induction regimens [119]. Another reason in favour of earlier use of moAbs is based on encouraging MRD rates achieved in most of the above-mentioned trials, even in highly pre-treated patients. Therefore, we may argue that introduction of these drugs in upfront regimens may facilitate the eradication of pathologic clones. On the other side, due to the short follow-up of patients enrolled in most recent trials incorporating the novel drugs, the duration of MRD-negativity is yet far from being predicted. In the case of patients relapsing after prior exposure to moAbs, rescue therapy could be challenging. This risk should be taken into consideration, especially for the elderly and for those relapsing with an aggressive clinical course: elderly patients may better tolerate moAbs than other drugs in advanced stage disease; for high-risk, rapidly evolving relapses, antibody therapy may be more feasible in a short time than other cellular treatments.

An accurate risk-stratification is necessary to help the identification of patients who might best benefit from the novel drugs from the beginning or in the earlier relapses. The definition of sensitive and specific molecular biomarkers at the baseline and the standardized evaluation of MRD are the next steps to be done to distinguish the different variants of the disease at the bench and to personalize the treatment at the bedside.

Therefore, the introduction of different antibody compounds represents another important milestone in the race for the cure of MM, and further translational research is required to advance toward the finish line.

## Figures and Tables

**Figure 1 biomolecules-12-01146-f001:**
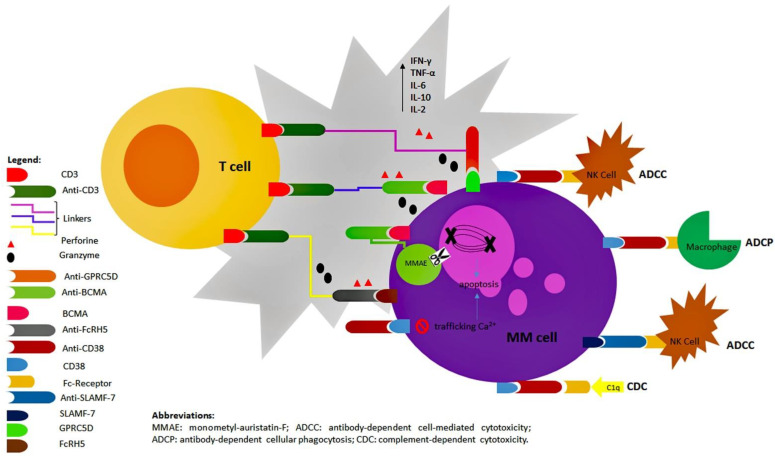
**Overview of mechanisms of action of mOabs.** MMAE: monometyl-auristatin-F; ADCC: antibody-dependent cell-mediated cytotoxicity; ADCP: antibody-dependent cellular phagocytosis; CDC: complement-dependent cytotoxicity.

**Figure 2 biomolecules-12-01146-f002:**
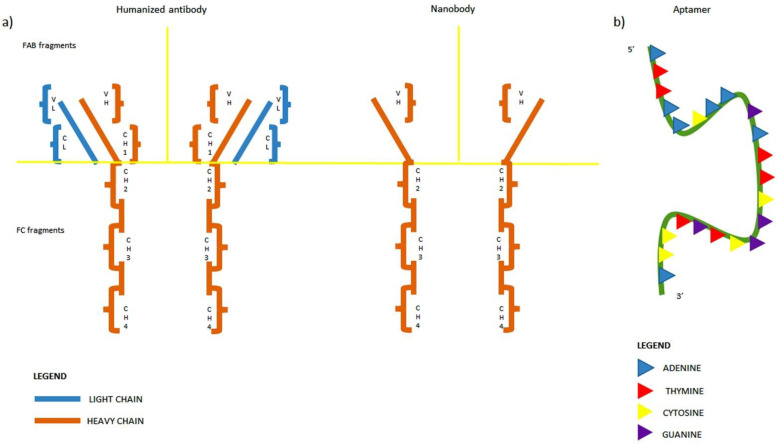
(**a**) **Comparison of antibody and nanobody structure**. In both cases, the yellow line separates the FAB fragments from the FC fragment. CH: constant heavy chain domain; VH: variable heavy chain domain; CL: constant heavy chain domain; VL: variable heavy chain domain. (**b**) Aptamer structure: single-stranded oligonucleotide with complementary sequence to the target.

**Table 1 biomolecules-12-01146-t001:** **Overview of “naked” moAbs with targets and clinical indications.** NDMM: newly diagnosed multiple myeloma; RRMM: relapsed-refractory multiple myeloma; pts: patients; dara: daratumumab; isa: isatuximab; fel: felzartamab; elo: elotuzumab; SA: single agent; V: velcade; M: melphalan; P: prednisone; R: lenalidomide; D: dexamethasone; T: thalidomide; P: pomalidomide; K: carfilzomib.

MECHANISM of ACTION	DRUG	SETTING
**ANTI-CD38**	DARATUMUMAB	NDMM: (a) transplant-ineligible pts: - Dara-VMP (ALCYONE) - Dara-RD (MAIA). (b) transplant-eligible pts: - Dara-VTD (CASSIOPEIA) - Dara-VRD (PERSEUS; GRIFFIN) RRMM - Dara-RD (POLLUX) - Dara-VD (CASTOR) - Dara-PD (APOLLO) - Dara-KD (CANDOR)
	ISATUXIMAB	RRMM - Isa-KD (IKEMA) - Isa-PD (ICARIA) - Isa SA - Isa-D
	FELZARTAMAB	RRMM - Fel SA - Fel-PD - Fel-RD
**ANTI-SLAMF-7**	ELOTUZUMAB	NDMM - Elo-RD (ELOQUENT-1) RRMM - Elo-RD (ELOQUENT-2) - Elo-PD (ELOQUENT-3)

**Table 2 biomolecules-12-01146-t002:** **Overview of “new” moAbs with targets and clinical indications.** NDMM: newly diagnosed multiple myeloma; RRMM: relapsed-refractory multiple myeloma; pts: patients; belamaf: belantamb-mafodotin; V: velcade; R: lenalidomide; D: dexamethasone; CPIs: check-point inhibitors; γSIs: γ-secretase inhibitors; SA: single agent.

ANTIBODY-DRUG CONJUGATES ANTI-BCMA	BELANTAMAB-MAFODOTIN	NDMM (a) Transplant-Eligible Pts: - Belamaf-VRD (NCT04802356) (b) Transplant-Ineligible Pts: - Belamaf-RD (NCT04808037) HRRMM - Belamaf SA (DREAMM-2) - Belamaf-VD/RD (DREAMM-6) - Belamaf-CPIs/ySIs (DREAMM-5)
	AMG224	highly pretreated RRMM
	MEDI2228	highly pretreated RRMM
	HDP-101	highly pretreated RRMM
**BISPECIFIC T-CELL ENGAGERS** **CD3xBCMA**	TECLISTAMAB	highly pretreated RRMM - SA (MAJESTEC-1)
	AMG420	highly pretreated RRMM - SA (NCT03836053)
	AMG701	highly pretreated RRMM - SA (NCT03287908)
	REGN5458	highly pretreated RRMM - SA (NCT03761108)
	CC-93269	highly pretreated RRMM - SA (NCT03486067)
	ELRANATAMAB	highly pretreated RRMM - SA (MAGNETISMM-3)
**BISPECIFIC T-CELL ENGAGERS** **CD3xOTHER TARGETS**	TALQUETAMAB (CD3xGPRC5D)	highly pretreated RRMM - SA (MONUMETAL-1)
	CEVOSTAMAB (CD3xFcRH5)	highly pretreated RRMM - SA (NCT03275103)

## Data Availability

Not applicable.

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
