# Peer review of "Race for the Cure: From the Oldest to the Newest Monoclonal Antibodies for Multiple Myeloma Treatment"

_biomolecules, 2022, doi:10.3390/biom12081146_

Round 1

Reviewer 1 Report

In this review the authors conduct an extensive overview on the use of monoclonal antibodies in multiple myeloma patients.

The paper is very well written, clear and very exhaustive. Here are a few observation:

-       Line 138: it is not very clear the link between the phase I trial of pomalidomide, daratumumab and dexamethasone with the CASTOR trial

-       Line 149: add reference

-       Since the use of Daratumumab in routine clinical practice has led to an increased incidence of infections, mainly pneumonia, could be of interest to expand a little the section 2.1.4. There is also a recent published Italian consensus-based position addressing this issue  (Girmenia et al, 2022).

-       Line 605: I would add that denosumab is approved for supportive care in MM

Author Response

We are very grateful for the relevant comments made to us.

Please see the attachment for the point-by-point responses.

Reviewer 2 Report

Overall, this is a well written overview on current antibody based immunotherapy in multiple myeloma.

I have only very few suggestions:

In line 368 I would be more cautious.  Antigen spreading and immunogenic cell death is still a matter of debate.  I would go for  … “potentially” inducing long-lasting adaptive (…)

Line 442 MEDI2228: I recommend to add the issue of photophobia. 58.5% of patients in the MTD cohort experienced photophobia; in 41.5% of patients this was grade 1 or 2, while it was grade 3 or 4 in 17.1%.

Line 557: Actually BCMA loss was also reported after bispecific antibodies.  Please see and reference the following paper Truger et al, Blood Advances 2021 PMID: 34471932, please also reference the paper on BCMA loss after CAR T cells PMID: 33619368

Author Response

(The authors gave the same response as above.)

Reviewer 3 Report

This review report is well written and comprehensively describes where we are now and the furture perspective in myeloma treatment.

I think it should be modified and reorganized a little to improve its readability. Several mistakes and typos should be corrected. 

1. Different font is used in the line 77 in the page 2.

2. I think pomalidmide must be bortezomib in the line 139.

3. Larger font is used in the line 152.

4. I don't think the paragraph from the line 227 which describes underestimation laboratory disease quantification should be included in the daratumumab side effect. It should be written in new section.

5. The authors describes that anti-SLAMF7 in combination works with both PI and IMiDs, but as far as I understand, elotuzmab did not work well in combination with PI bortezomib, therefore combinations with IMiDs (lenalidomide and pomalidmide) have been studied and produced good result. 

6. I think trials of elotuzumab for newly diagnosed myeloma have been failed.

7. Both isatuximab alone and isatuximab+dexamethasone also showed efficacy for relapse refractory myeloma, so those regimens should be included in article and table 1. 

8. 400μg/die must be 400μg/kg in the line 498.

9. In the section 3.2.3) BiTE non targeting BCMA, the paragraph titles such as GPRC5D and FcRH5 should be added at the head of the paragraphs for better readability. 

10. Large fonts were used from the line 597 to 607.

11. Section 4 is too long. It should be divided two or three small sections. For example, nanobody section and radionucleotide section, aptamer sections. And it would very nice for readers to understand if figures well exlaining those nanoparticles are added.  

12. The authors mentions reduction of manufacturing cost in the line 646, but is there any evidence about its cost?

Author Response

(The authors gave the same response as above.)
